# Antioxidant Network Based on Sulfonated Polyhydroxyalkanoate and Tannic Acid Derivative

**DOI:** 10.3390/bioengineering8010009

**Published:** 2021-01-08

**Authors:** Laura Brelle, Estelle Renard, Valerie Langlois

**Affiliations:** Univ Paris Est Creteil, CNRS, ICMPE, UMR 7182, 2 rue Henri Dunant, 94320 Thiais, France; brelle@icmpe.cnrs.fr (L.B.); renard@icmpe.cnrs.fr (E.R.)

**Keywords:** polyhydroxyalkanoate, PHOU, water soluble PHA, network, tannic acid

## Abstract

A novel generation of gels based on medium chain length poly(3-hydroxyalkanoate)s, *mcl*-PHAs, were developed by using ionic interactions. First, water soluble *mcl*-PHAs containing sulfonate groups were obtained by thiol-ene reaction in the presence of sodium-3-mercapto-1-ethanesulfonate. Anionic PHAs were physically crosslinked by divalent inorganic cations Ca^2+^, Ba^2+^, Mg^2+^ or by ammonium derivatives of gallic acid GA-N(CH_3_)_3_^+^ or tannic acid TA-N(CH_3_)_3_^+^. The ammonium derivatives were designed through the chemical modification of gallic acid GA or tannic acid TA with glycidyl trimethyl ammonium chloride (GTMA). The results clearly demonstrated that the formation of the networks depends on the nature of the cations. A low viscoelastic network having an elastic around 40 Pa is formed in the presence of Ca^2+^. Although the gel formation is not possible in the presence of GA-N(CH_3_)_3_^+^, the mechanical properties increased in the presence of TA-N(CH_3_)_3_^+^ with an elastic modulus G’ around 4200 Pa. The PHOSO_3_^−^/TA-N(CH_3_)_3_^+^ gels having antioxidant activity, due to the presence of tannic acid, remained stable for at least 5 months. Thus, the stability of these novel networks based on PHA encourage their use in the development of active biomaterials.

## 1. Introduction

Poly(3-hydroxyalkanoate)s (PHAs) are natural polymers produced by microorganisms as carbon and energy storage compounds when they are placed under special growth conditions. PHAs are naturally occurring aliphatic polyesters that have a high structural diversity related to the alkyl pendant group [1,2,3]. The length of this side chain group is decisive for their final thermal and mechanical properties. PHAs can be divided into three main types depending on the number of carbon atoms in the monomer unit: short-chain-length PHAs (*scl*-PHA), with monomers consisting of 3–5 carbon atoms, medium-chain-length PHAs (*mcl*-PHA), with monomers consisting of 6–14 carbon atoms, and long chain length PHAs (*lcl*-PHA), which are obtained from long chain fatty acids with more than 14 carbon atoms. The former display physical properties associated with highly crystalline thermoplastics while the latter generally possess low crystallinity and elastomeric character. Due to their biodegradability [4,5] and biocompatibility, they constitute a very promising group of natural biomaterials for various biomedical applications including drug delivery and tissue engineering [6,7,8,9]. 

In order to enlarge the potential in biomedical applications, chemical modifications of *mcl*-PHAs have been performed. One of the first studies concerning the chemical modification of PHAs started with the production of poly(3-hydroxyoctanoate-*co*-3-hydroxyundecenoate) (PHOU) that contained unsaturated groups in the side chains [10,11]. The chemical modifications of the unsaturated groups of PHOU have already been reported [12,13,14,15,16,17]. The introduction of polar groups, such as hydroxyl, carboxylic and ammonium groups, makes it possible to modify the hydrophilic/hydrophobic balance of the polymer. Amphiphilic PHAs-PEG copolymers that were synthesized by esterification [18,19,20] and by click chemistry [21,22,23,24] have shown the ability to self-associate in aqueous medium to form micelles, vesicular structures or polymersomes. Water soluble PHAs obtained by introducing PEG [18], ammonium groups [25], or sulfonate functions [26,27] retained their biocompatible and non-cytotoxic character and the presence of PEG and propyleneglycol (PPG) segments provided hydrophilicity, blood compatibility [28], and thermo-sensitivity respectively [29,30]. 

Networks based on PHAs were prepared by photoactivated reaction in the presence of PEG-monoacrylate [28] or PEG-diacryloyl [31]. Chen et al. [32] recently reported PHA gel via photo-crosslinking based on a thiol-ene reaction with PEG-dithiol as a photo-cross-linker. Although some structures from chemically cross-linked PHAs have already been described in the literature, there is to our knowledge no study on the gels with PHAs bearing sulfonate groups. These novel materials should constitute an interesting support in the biomedical field because of the biological interactions of the sulfonate groups due to their anticoagulant, antiangiogenic, and antimitogenic properties. These active anionic sites are known to interact with proteins, such as fibronectin [33,34,35,36].

In our paper, we describe the synthesis of PHA sulfonate, PHOSO_3_^−^, by a photoactivated thiol-ene process in the presence of sodium-2-mercapto-1-ethanesulfonate and the elaboration of crosslinked structures by combination with bivalent cations or organic polycationic molecules (Figure 1).

Among the bivalent cations that have the capability of forming a gel by ionic interactions with anionic polymers, Ca^2+^, Ba^2+^, and Mg^2+^ present interesting properties in biomedical applications [37,38,39]. In particular, the gelation of an alginate solution with a multivalent ion of opposite charge such as Ca^2+^ allows to obtain a resistant and biocompatible hydrogel.

To increase the numbers of ionic interactions, we report here the synthesis of ammonium derivatives of gallic acid and tannic acid. Gallic acid (GA) is a natural tri-hydroxybenzoic acid extracted from plants and evaluated as an antioxidant, antibacterial, and antiviral compound [40,41]. Tannic acid (TA) is a kind of natural polyphenol that is commonly found in high herbaceous and woody plants that is a well-known bio-based molecule for its antioxidant and antibacterial activity [42,43,44]. Moreover, polyphenols due to their cyclic structure and the presence of a large content of hydroxyl groups are good candidates for the development of stable physical gels and offer the possibility to be easily modifiable through these accessible functions. Tannic acid has already shown the ability to make gels or hydrogels with good mechanical structures by different interactions such as supramolecular interactions by hydrogen bonds [45], coordination with transition metals [46,47], and a self-assembly method by redox reaction [48,49]. A series of biobased UV-curable antibacterial resins were synthesized through modifying tannic acid with glycidyl methacrylate (GMA) [50]. In this study, ammonium groups were introduced on GA and TA through reaction with glycidyl trimethyl ammonium chloride (GTMA). The presence of ammonium groups on GA and TA increases both the number of cationic charges compared to divalent cations and the rigidity by the presence of the aromatic rings. Gels were formed by ionic interactions of the oppositely charged polyelectrolytes PHOSO_3_^−^ and Ca^2+^, Ba^2+^ and Mg^2+^, and GA-N(CH_3_)_3_^+^ and TA-N(CH_3_)_3_^+^. The viscoelastic properties of the gels and their antioxidant activity were studied.

## 2. Materials and Methods 

### 2.1. Materials

Poly(3-hydroxyoctanoate-co-3-hydroxyundecenoate) with 33% double bonds PHOU was provided from the Swiss Federal Laboratory for Materials Testing and Research (EMPA, Switzerland, Mn = 40,000 g/mol, Polydispersity index 1.7). The different comonomers are randomly distributed. CaCl_2_, BaCl_2,_ MgCl_2_, and Sodium 2-mercapto-1-ethanesulfonate (98%) were purchased from Aldrich. Tannic acid was purchased from Alfa Aesar. Tetrahydrofuran (THF), dimethylsulfoxide (DMSO), methanol, and ethanol absolute anhydrous were purchased from Carlo Erba. Gallic acid, 2,2-dimethoxy-2-phenylacetophenone (DMPA, Irgacure 651), glycidyltrimethyl ammonium (GTMA), triphenylphosphine (TPP), and methyl hydroxyquinone (MEHQ), were supplied from Sigma-Aldrich and were used without prior purification. Butylacetate was supplied from Merck. Tannic acid was purchased from Alfa Aesar.

### 2.2. Synthesis of PHO Sulfonate, PHOSO_3_^−^

A total of 0.2 g PHOU was solubilized in 12 mL THF under agitation at room temperature. Five molar equivalents (relative to the double bonds of PHOU) of sodium 2-mercapto-1-ethanesulfonate (98%) and 0.5 eq. molar DMPA were solubilized in 18 mL of a MeOH/THF (2/1 *v*/*v*) mixture. The mixture placed at 11 cm was irradiated for 300 s at room temperature with a mercury xenon (180 Mw·cm^−2^) Lightning cure LC8 (L8251) lamp from Hamamatsu coupled with a flexible light guide. During the entire irradiation process the mixture was kept under stirring. After irradiation, 25 mL DMSO was added. The resulting product was dialyzed in a 1000 Da cut-off dialysis tube against distilled water, which was changed three times daily for 4 d.

### 2.3. Synthesis of Trimethyl Ammonium Gallic Acid: GA-N(CH_3_)_3_^+^

In a two-neck round-bottom flask, 2.125 g of gallic acid, 4.996 × 10^−2^ moles of GTMA, 0.1456 g of triphenylphosphine (TPP) and 0.0097 g of methyl hydroxyquinone (MEHQ) were solubilized in 5 mL of anhydrous ethanol and butyl acetate 1:2 (*v*:*v*), under an inert atmosphere. The mixture was stirred at 100 °C for 48 h. After 48 h, the mixture was precipitated in acetone to remove solvent and then purified by flash column chromatography using methanol as an eluent.

### 2.4. Synthesis of Trimethyl Ammonium Tannic Acid: TA-N(CH_3_)_3_^+^

In a flask, 4.25 g of tannic acid, 0.25 moles of GTMA, 0.09 g of triphenylphosphine (TPP) and 0.0075 g of methyl hydroxyquinone (MEHQ) were solubilized in 10 mL of an anhydrous ethanol and butyl acetate 1:2 (*v*:*v*) mixture, under an inert atmosphere. The mixture was stirred at 100 °C for 48 h. Then it was solubilized in an H_2_O/ethanol mixture (25:75) and centrifuged. The precipitate containing the TA-N(CH_3_)_3_^+^ was recovered and dried in a vacuum chamber at 40 °C for one night. The molar mass of the monomeric unit of the polymer is M_0_ = (0.67 × 142) + (0.33 × 323) = 201 g·mol^−1^. 

### 2.5. Elaboration of Network Based on PHOSO_3_^−^

PHOSO_3_^−^/Ca^2+^. Networks were prepared according to the following protocol. A total of 0.05 mL of CaCl_2_ 0.2 M was added to 0.25 mL of PHOSO_3_^−^ with a concentration of 27 g·L^−1^. The mixture was vortexed for 30 s allowing the instantaneous formation of the network. 

PHOSO_3_^−^/TA-N(CH_3_)_3_^+^. A total of 0.2 mL of PHOSO_3_^−^ with a concentration of 96 g·L^−1^ and 0.01 g of TA-N(CH_3_)_3_^+^ were introduced into a haemolysis tube. After an agitation with a vortex, a sticky, brown gel was formed. 

### 2.6. DPPH Test

The radical scavenging activity (RSA) of the networks was determined by using 2,2-diphenyl-1-picrylhydrazyl. A total of 37 mg of the PHOSO_3_^−^/TA-N(CH_3_)_3_^+^ were immersed in 3 mL of 0.1 mM of DPPH solution in methanol in the dark for 60 min at room temperature. The RSA was measured by using a Varian Cary 50 Bio UV-Visible spectrophotometer controlled by the CaryWinUV software. RSA was therefore determined by monitoring the decrease of the absorbance at 517 nm (Equation (1)): RSA (%) = ((A_ref_ − A_s_)/A_ref_) × 100 (1)
where A_ref_ corresponds to the absorbance of the 0.1 mM of DPPH solution without a sample and A_s_ corresponds to the absorbance of the 0.1 mM solution of DPPH with 37 mg samples of tannic derivatives.

### 2.7. Characterization

The molar mass of the PHO_(67)_U_(33)_ was determined by Size Exclusion Chromatography in THF using a Schimadzu LC-10AD pump with two Shodex GPC K-805L columns (5 µm Mixte-C) at a concentration of 10 mg·mL^−1^. A Wyatt Technology Optilab Rex interferometric refractometer (Toulouse, France) was used as a detector, and low polydispersity index polystyrene standards (3 × 10^4^–2 × 10^6^ g/mol) were used for PHOU analysis. ^1^H NMR spectra (400 MHz) were performed with a Bruker AV 400M spectrometer (Wissembourg, France). In a disposable spectrophotometer cell, 10 µL of CaCl_2_ (0.2 M) was added to 0.250 mL of PHOSO_3_^−^ at 37 g·L^−1^, in a constant manner. After manual stirring, the transmittance was measured at 600 nm with a UV-vis Cary50 Bio Varian UV−visible spectrometer supplied by Varian (Agilent, Les Ulys, France) recording controlled by Cary Win UV software in the range 250–800 nm. Rheological properties were determined with a hybrid rheometer (DISCOVERY HR-2) (Guyancourt, France) using a cone and plate geometry (20 mm, 1°). Rheological tests were performed at 25 °C (room temperature) and the Peltier temperature control system was used to keep the temperature constant throughout the analysis. The evaporation of the water contained in the 3D network was limited by the presence of a solvent trap. For PHOSO_3_^−^/M^2+^ networks, the elastic modulus (G’) and viscous modulus (G”) were plotted against frequency from 0.1 et 100 rad·s^−1^ for an applied strain at 20% on 0.350 mL of a 3D network. In addition, for the PHOSO_3_^−^/TA-N(CH_3_)_3_^+^ network frequency sweep tests were done between 0.1 and 100 Hz for an applied strain at 1%. These measurements were carried out three times to ensure the repeatability of the results and the viability of the tests. 

## 3. Results and Discussion

### 3.1. Synthesis of Poly(3-Hydroxyalkanoate) Sulfonate, PHOSO_3_^−^, Ammonium Derivatives of Gallic Acid GA-N(CH_3_)_3_^+^ and Tannic Acid, TA-N(CH_3_)_3_^+^

Poly(3-hydroxyoctanoate-co-3-hydroxyundecenoate) PHOU is a hydrophobic polyester composed of long alkyl side chains and lateral unsaturated groups. Sulfonate groups were grafted to the terminal double bonds of the PHOU, to both provide an amphiphilic character to the PHA and to induce non-covalent binding of SO_3_^−^ ions to cationic species to form ionic crosslinking (Figure 1). 

Poly(3-hydroxyoctanoate-co-3-hydroxyundecenoate), PHOU was first characterized by ^1^H NMR (Figure 2) to determine the percentage of terminal unsaturation by integrating protons corresponding to the CH peak (2) at 5.1 ppm, and the signal relating to the terminal alkene group of side chain (7) at 5.7 ppm. Sodium-3-mercapto-1-ethanesulfonate was grafted under photochemical activation, using 5 molar equivalents of sulfonate groups. After purification by dialysis, the polymers were dried by freeze-drying and then analyzed by ^1^H NMR. The presence of the signals relating to the two CH_2_ methylene (c, d), which are characteristic of sulfonate groups, appeared at 2.66 ppm and the total disappearance of the peaks relating to the terminal alkene group of side chains (7) and (8) indicated the success of the formation of PHOSO_3_^−^. We previously showed that no chain scission occurred during this thiol-ene reaction and the PHOSO_3_^−^ is water soluble [26,27].

Ammonium derivative of gallic acid GA-N(CH_3_)_3_^+^ and tannic acid TA-N(CH_3_)_3_^+^ were synthesized through the reaction with glycidyl trimethylammonium chloride GTMA (Figure 3). ^1^ H NMR spectra (Figure 4) showed that GTMA was grafted to phenol groups of the TA due to the disappearance of the peak at 3.1 ppm (a) which correspond to the epoxide groups of GTMA and the appearance of peaks at 3.5–4.6 ppm, which correspond to the -CH_2_-CH(OH)-CH_2_-N(CH_3_)^+^ units (a’, b’, c’). The number of ammonium groups N(CH_3_)_3_^+^ was calculated from the area ratio of the peaks at 6.7–7.5 ppm (*I*_1,2_), which corresponds to the protons on the aromatic group of tannic acid, and the peak at 3.2 ppm (*I_d_*), which are characteristic of the protons on the ammonium groups (Equation (2)). The conversion of the grafting reaction is defined by Equation (3).
(2)$$N{\left({\mathrm{CH}}_{3}\right)}_{3}{}^{+}=\frac{I_{1,2}\;\times 20}{I_{d}\;\times 9}$$
(3)$$\mathrm{Conversion}\;\left(\%\right)=\frac{N{\left({\mathrm{CH}}_{3}\right)}_{3}{}^{+}}{25}\;\times 100$$

The substitution degree of TA was calculated as around 76% (which corresponds to 19 N^+^ per molecule) and TA-N(CH_3_)_3_^+^ is totally soluble in water. The substitution reaction of GA determined by NMR is quantitative. The degree of substitution was calculated as the ratio of the integration of the peak (d) at 3.3 ppm to the number of hydrogens carried by the nitrogen atom (N^+^), i.e., nine. Integration of peaks b and c confirmed that there were 4 N^+^ per molecule. As a consequence, we can further study the influence of the numbers of ammonium groups on the gel formation.

### 3.2. Effects of the Nature of Cations on the Formation of Gels

Networks were spontaneously formed after a few seconds of mixing PHOSO_3_^−^ and cationic species. The influence of the nature of divalent cations as Mg^2+^, Ba^2+^, and Ca^2+^ was first studied. In the presence of Mg^2+^ no gel formation was observed regardless of the concentration. Mg^2+^ therefore does not have sufficient interaction energy and attractive power [51]. The Ba^2+^ did not allow the formation of a gel and gave rise to a precipitate even at very low concentrations. This result may be due to the bigger size of the barium atom nucleus combined with an insufficient proportion of sulfonate functions [52]. PHOSO_3_^−^, which is completely soluble in water (27 g·L^−1^; 1.1 × 10^−5^ moles), instantly forms a network in the presence of a sufficient molar concentration of Ca^2+^. The variation of transmittance as a function of the number of moles of Ca^2+^ showed three well-defined zones (Figure 5). A first zone is defined between 0 to 1.5 × 10^−5^ moles, showing that the polymer remains in solution. Between 1.5 × 10^−5^ and 2.5 × 10^−5^ moles, the transmittance drops sharply from 98% to 20%, respectively, attesting to the gel formation. The addition of Ca^2+^ beyond this concentration does not modify the transmittance, which remains constant at 20%. The presence of precipitate is then observed when the moles of salt are greater than 2.6 × 10^−5^ moles.

The gel formation conditions were studied next in the presence of Ca^2+^, gallic acid and tannic acid derivatives. Under our conditions, no gel was formed in the presence of GA-N(CH_3_)_3_^+^ whereas gels formed very easily in the presence of TA-N(CH_3_)_3_^+^, which allows us to affirm that the number of positive charges is very important during gel formation. Therefore, in order to be able to compare the conditions of gel formation in the presence of Ca^2+^ and the TA-N(CH_3_)_3_^+^ derivative, Figure 6 shows the different zones observed for the gels as a function of the number of positive and negative charges. To perform this study we added increasing volumes of positive charge solution (Ca^2+^ or TA-N(CH_3_)_3_^+^) from the same initial solution. The results obtained are very different because there are three zones in the case of Ca^2+^ (liquid, gel and precipitate), whereas there are only two zones for TA-N(CH_3_)_3_^+^ (liquid and gel). In the presence of Ca^2+^, the gels were formed in the stoichiometric conditions of the number of moles of positive and negative charges. On the other hand, in the presence of TA-N(CH_3_)_3_^+^, the experimental conditions were different because it is enough that the number of moles of sulfonate groups and positive charges are greater than 2.296 × 10^−5^ moles and 0.483 × 10^−5^ moles, respectively.

### 3.3. Structure of Networks and Antioxidant Properties

The mechanical properties of the PHOSO_3_^−^/Ca^2+^ networks are strongly dependent of the polymer and CaCl_2_ concentration (Table 1). Dynamic viscoelastic measurement results show that at the higher frequency region G’ is larger than G”, showing that these gels present predominantly elastic properties. The modulus G’ depends on the concentration of CaCl_2_ and increases linearly with the concentration of PHOSO_3_^−^, which shows the influence of hydrophobic interactions between chains and undoubtedly the presence of entanglements. This effect is superimposed on the ionic bonds between the polymer and Ca^2+^. However, the elastic modulus obtained remains low and does not exceed 40 Pa. Although the network keeps its structure for a week, the addition of water led to the deterioration of the network. The low percentage of SO_3_^−^ groups (33%) did not allow the formation of enough ionic interactions with Ca^2+^. The use of other calcium salts such as CaCO_3_ was studied, but its low solubility in water did not allow the formation of the network, or even CaSO_4_·2H_2_O in the presence of Na_2_HPO_4_, which is often used as a gelation accelerator [38].

Although the formation of a tannic acid-based network often requires a certain pH value necessary to have the phenolate groups [44,45,46], the advantage of our system is that there is no pH-dependence of the gel formation. In the physiological medium, the tannic acid derivative TA-N(CH_3_)_3_^+^ and the sulfonate PHA always have quaternary amine and sulfonate groups, respectively. The PHOSO_3_^−^/TA-N(CH_3_)_3_^+^ network has better mechanical properties than the gel obtained in the presence of Ca^2+^ (Figure 7). Indeed, the viscoelastic modulus at 100 Hz was much larger than those of the PHOSO_3_^−^/Ca^2+^ network. The elastic modulus of the PHOSO_3_^−^/TA-N(CH_3_)_3_^+^ network was stable even after 5 months. This was due to the presence of polyphenolic moieties that gave the network a very high stability and remarkable mechanical properties despite the low proportion of SO_3_^−^. Unlike the PHOSO_3_^−^/Ca^2+^ network, the addition of water to the PHOSO_3_^−^/TA-N(CH_3_)_3_^+^ network did not disrupt the structure of the network, and the network structure was maintained even after the rheological studies. Moreover, the formation of the network was a completely reversible process even after freeze-drying without any change of the viscoelastic modulus.

The antioxidant capacity was assessed using the DPPH radical scavenging assay. This method consists in performing a reduction of the DPPH radical to its non-radical form in the presence of tannic acid, a hydrogen donating compound (Figure 8). The solution containing DPPH has a purple color, corresponding to a UV-visible absorbance at 517 nm. In the presence of TA, TA-N(CH_3_)_3_^+^, and PHOSO_3_^−^/TA-N(CH_3_)_3_^+^, the solutions became yellow and presented significant antiradical activity with a radical scavenging activity (% RSA) superior to 82%.

## 4. Conclusions

A copolymer based on unsaturated PHAs was functionalized with the polar SO_3_^−^ sulfonate groups to promote, on the one hand, its solubility in water [26,27], and on the other hand, ionic interactions with cations. An efficient and reproducible method for the synthesis of copolymer of water soluble PHOSO_3_^−^ was developed by thiol-ene reaction in the presence of sodium-3-mercapto-1-ethanesulfonate. This copolymer was further used for the preparation of network by ions’ interaction with different cations, namely Ca^2+^, Ba^2+^, and Mg^2+^. Among them, only Ca^2+^ presented the ability to form a gel in the well-defined contents. To reinforce the mechanical properties and stability of the gels, a polycationic derivative of tannic acid was synthesized by reaction with glycidyl trimethyl ammonium chloride (GTMA). The presence of aromatic groups and cationic charges improved the elastic modulus from 40 Pa obtained with Ca^2+^ to 4200 Pa in the presence of TA-N(CH_3_)_3_^+^. The antioxidant networks based on PHOSO_3_^−^/TA-N(CH_3_)_3_^+^ were stable for 5 months in a buffered physiological environment and constitute a new generation of soft biomaterials.

## Figures and Tables

**Figure 1 bioengineering-08-00009-f001:**
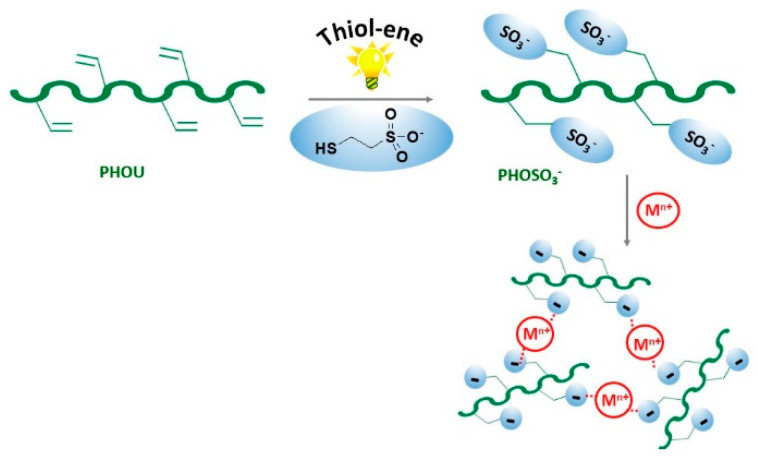
Gelation process of PHOSO_3_^−^ induced by cationic species as Ca^2+^, Mg^2+^, Ba^2+^, and gallic and tannic acid derivatives.

**Figure 2 bioengineering-08-00009-f002:**
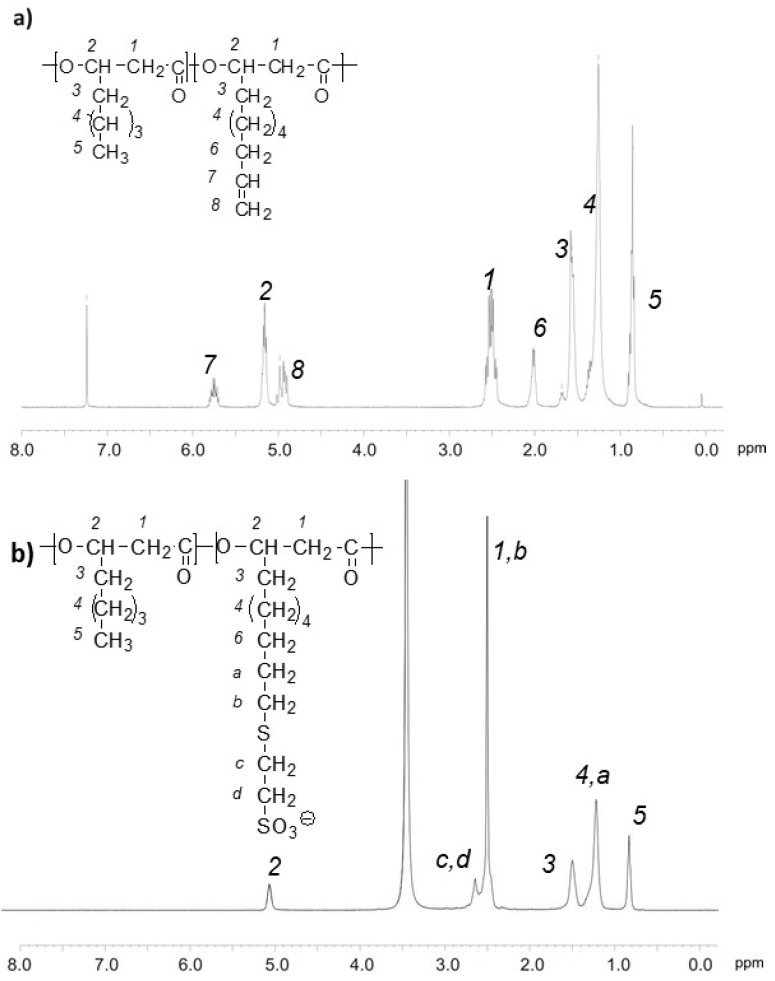
^1^H NMR spectra: (**a**) PHOU in CDCl_3_; (**b**) PHOSO_3_^−^ in dimethyl sulfoxide, DMSO-d6.

**Figure 3 bioengineering-08-00009-f003:**
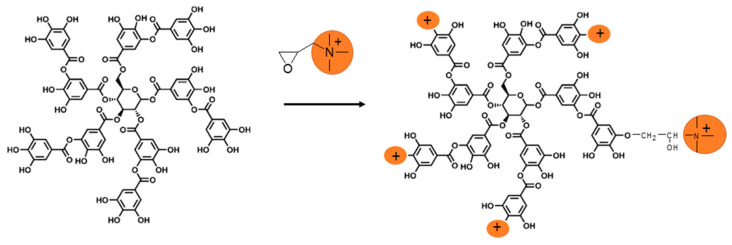
Schematic representation of TA-N(CH_3_)_3_^+^ by chemical modification of tannic acid in the presence of GTMA.

**Figure 4 bioengineering-08-00009-f004:**
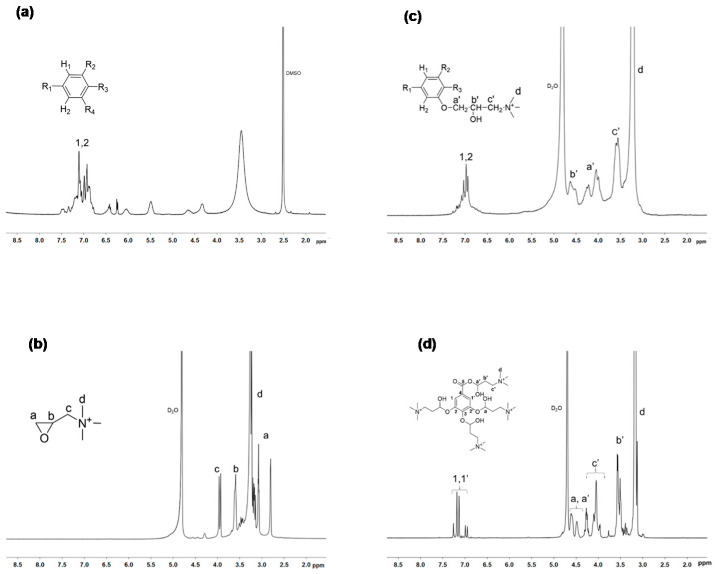
^1^H NMR spectra of (**a**) tannic acid in DMSO, (**b**) GTMA in D_2_O, (**c**) TA- N(CH_3_)_3_^+^ in D_2_O, and (**d**) GA-N(CH_3_)_3_^+^ in D_2_O.

**Figure 5 bioengineering-08-00009-f005:**
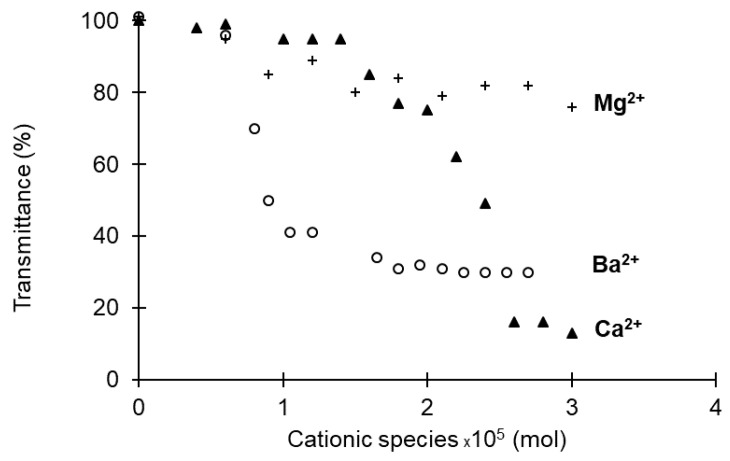
Effect of the cationic species on the network formation with a PHOSO_3_^−^ (concentration of 27 g·L^−1^).

**Figure 6 bioengineering-08-00009-f006:**
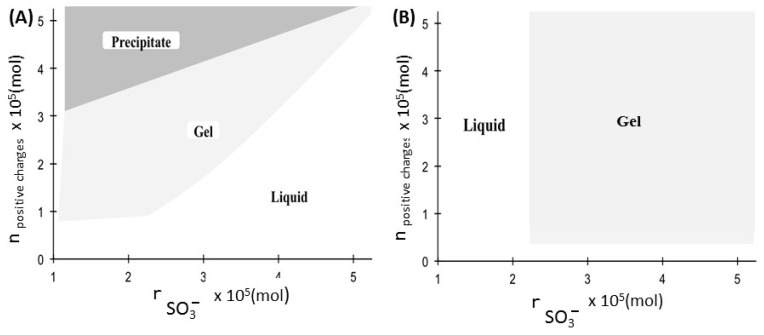
Schematic representation of the effect of the number of positive charges for Ca^2+^ (**A**) or TA-N(CH_3_)_3_^+^ (**B**) as a function of the amount of SO_3_^−^ in the gel formation.

**Figure 7 bioengineering-08-00009-f007:**
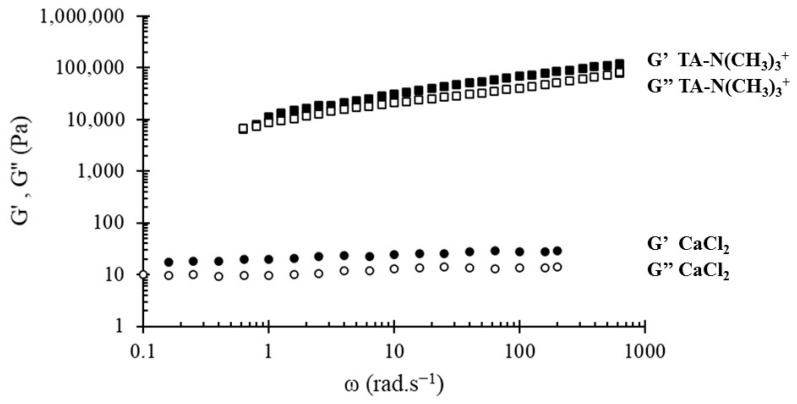
Frequency dependences of the storage modulus G’ and loss modulus G’’ of PHOSO_3_^−^ at 96 g·L^−1^ with CaCl_2_ and TA-N(CH_3_)_3_^+^.

**Figure 8 bioengineering-08-00009-f008:**
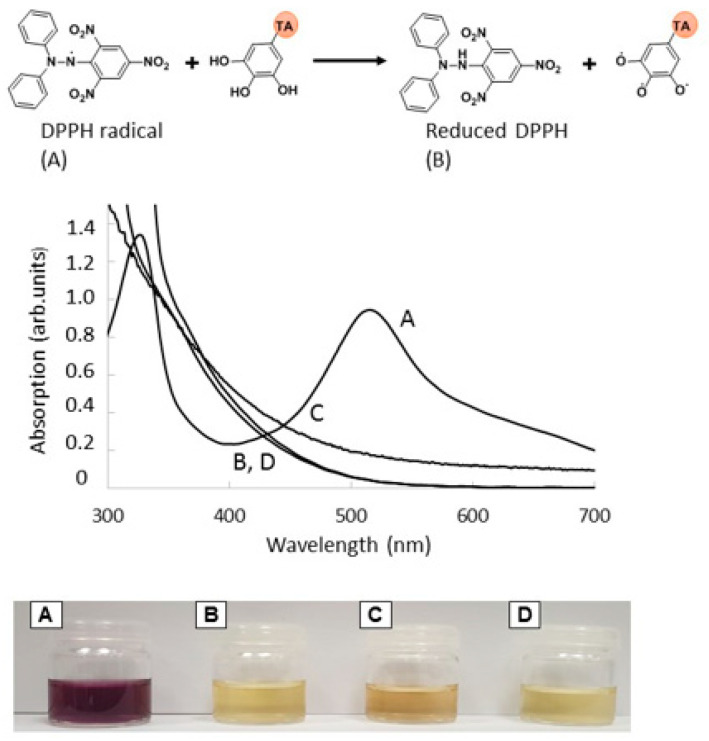
Absorption UV spectra in the presence of DPPH (**A**) and reduced DPPH after reaction with TA (**B**), TA-N(CH_3_)_3_^+^ (**C**), and PHOSO_3_^−^/TA-N(CH_3_)_3_^+^ (**D**).

**Table 1 bioengineering-08-00009-t001:** Influence of PHO-SO_3_^−^ concentration on gel mechanical properties of PHOSO_3_^−^/Ca^2+^ networks at 10 Hz.

[PHO-SO_3_^−^] (g/L)	n_SO3_^−^ ×10^5^	n_Ca_^2+^ ×10^5^	n Positive Charges ×10^5^	G’ (Pa)	G” (Pa)
47	1.93	1.4	2.8	4.01	3.93
59	2.42	1.5	3.0	14.62	11.26
68	2.79	1.4	2.8	24.20	12.78
71	2.91	1.8	3.6	38.38	24.92

## Data Availability

Not applicable.

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
