# Peer review of "Antioxidant Network Based on Sulfonated Polyhydroxyalkanoate and Tannic Acid Derivative"

_bioengineering, 2021, doi:10.3390/bioengineering8010009_

Round 1

Reviewer 1 Report

The controllability of the biodegradation process and a wide range of mechanical characteristics make it possible to put forward PHAs, group of natural polymeric materials,over materials,based on copolymers of lactic and glycolic acids, as well as polydioxanone and polycaprolactone, i.e. basic biodegradable materials in medical practice. PHA are unique materials for tissue engineering of the future, as reflected in numerous publications, which have to be emphasised in this work. Namely, in the introduction, one should mention in numbres the diversity of physical and mechanical characteristics of natural samples of PHA of various monomeric composition, with reference to Volova's articles (*), e.g. modulus of elasticity, etc., as well as, accordingly, differences in the biodegradation of materials, which is the basis for using products made from them in bioengineering (**).

*

1) Natalia Zhila, Ekaterina Shishatskaya,

Properties of PHA bi-, ter-, and quarter-polymers containing 4-hydroxybutyrate monomer units,

International Journal of Biological Macromolecules,

Volume 111,

2018,

Pages 1019-1026,

ISSN 0141-8130,

https://doi.org/10.1016/j.ijbiomac.2018.01.130

**

2) Volova, T. G., Prudnikova, S. V., Vinogradova, O. N., Syrvacheva, D. A., & Shishatskaya, E. I. (2017),

Microbial degradation of polyhydroxyalkanoates with different chemical compositions and their biodegradability. 

Microbial ecology, 73(2), 353-367

DOI 10.1007/s00248-016-0852-3

Author Response

Dear Reviewer

Thank you for your positive comments.

We improved the introduction as it was suggested and we added a reference to Volova's article in the manuscript. 

Volova, T. G., Prudnikova, S. V., Vinogradova, O. N., Syrvacheva, D. A., & Shishatskaya, E. I. (2017),Microbial degradation of polyhydroxyalkanoates with different chemical compositions and their biodegradability, Microbial ecology, 73(2), 353-367

Reviewer 2 Report

Renewable Antioxydant Network based on 3 Sulfonated Polyhydroxyalkanoate and Tannic acid 4 Derivative is an interesting article about synthesis of gels from renewable poly(3-hydroxyalkanoate).

The objective of the authors were the modification and characterization of antioxidants from Poly(3-hydroxyoctanoate-co-3-hydroxyundecenoate) with 33% double bonds (PHOU), however, some of the characterization reports in Materials and Methods section have not been shown and others were not sufficiently to necessary identification.

The molar mass determination of the PHO67U33 which author write as determined by SEC not was reported, however the molar mass and dispersion are extremely relevant in the study of the rheological behavior.

The 1H NMR characterization of the of the PHO67U33 was incompleted, since this type of copolymer with 33% moles of insatured comonomer introduce substantially changes in physic and chemistry behaviour in dependence of the comonomer distribution in the polymer chains and should be influence on the gel structure.

FTIR/ATR was mention as one of the characterization techniques without authors report results, drawing particular attention to the use of ATR in this study.

Another aspect is the authors emphasis (Conclusions) about water solubility of the PHOSO3- polymer without showing results about it study.

Author Response

Dear Reviewer,

Thank you for your positive comments that help us to improve the quality of the manuscript. We provide a point-by-point response and we modified the manuscript.

The molar mass determination of the PHO67U33 which author write as determined by SEC not was reported, however the molar mass and dispersion are extremely relevant in the study of the rheological behavior.

The 1H NMR characterization of the of the PHO67U33 was incompleted, since this type of copolymer with 33% moles of insatured comonomer introduce substantially changes in physic and chemistry behaviour in dependence of the comonomer distribution in the polymer chains and should be influence on the gel structure.

FTIR/ATR was mention as one of the characterization techniques without authors report results, drawing particular attention to the use of ATR in this study.

Response : Poly(3-hydroxyoctanoate-co-3-hydroxyundecenoate) with 33% double bonds PHOU was provided from the Swiss Federal Laboratory for Materials Testing and Research (EMPA, Switzerland, Mn= 40000 g/mol, Polydispersity index 1.7). In this bacterial polyesters, the different comonomers are randomly distributed. 

We removed the FTIR/ATR measurements.

2- Another aspect is the authors emphasis (Conclusions) about water solubility of the PHOSO3- polymer without showing results about it study. 

We previously showed that PHAs bearing SO3- were water soluble [26-27].

26. Modjinou, T.; Lemechko, P.; Babinot, J.; Versace, DL, Langlois, V., Renard, E. Poly(3-hydroxyalkanoate) sulfonate: From nanopartices toward water soluble polyesters, Polym. J. 2015,68,471-479.

27. Jain-Beuguel, C.; Li, X.; Houel-Renault, L.; Modjinou, T.; Simon-Colin, C.; Gref, R.; Renard, E.; Langlois, V. Water-Soluble Poly(3-hydroxyalkanoate) Sulfonate: Versatile Biomaterials Used as Coatings for Highly Porous Nano-Metal Organic Framework, Biomacromolecules 2019,20,3324-3332.

Reviewer 3 Report

This manuscript describes the synthesis of a water-soluble, sulphonated derivative of a medium-chain length polyhydroxyalkanoate (PHA). In the presence of certain cations, the sulphonated polymer forms gels with different mechanical properties. Inclusion of a derivative of tannic acid leads to a gel with antioxidant properties.

The work is properly carried out and the results are convincing. Nevertheless, there are several issues that must be solved for the manuscript to be acceptable.

1.- The presentation of results is, overall, very confusing, and it takes a long time and effort for the reader to get the necessary insight. For instance, the molecular mass of the sulphonated polymer is not given. This makes it very difficult to translate concentrations in g/L into moles of negative charges. An explanation on this procedure should be detailed. Moreover, some quantities are shown in moles, other in g/L, grams... while all these are useful to be maintained, for instance to easily reproduce the experiments, at least the number of moles should be always added to the text as well.

2.- The Experimental Procedures section does not always reflect the experiments shown in the text (e.g. Figs 5 or 6), but only one condition for gel formation is indicated.

3.- The degree of substitution of tannic acid by amine groups is said to be 88% and corresponding to 14 nitrogens with positive charge. It would be useful to explain in more detail the derivatization reaction and subsequent calculations made by the authors. For instance, upon inspection of Figure 3, it might be thought that only those OH groups in the "terminal" aryl rings are able to react with the epoxyde. If this were the case, there are 15 of these groups (3 per each terminal ring), of which the 88% accounts for 13, not 14 nitrogens added. Also, this would mean that the rest of OH groups are not reactive. Is this anything that has been described before?

4.- Pages 6-7, lines 203-209, and Figure 5. I have estimated that the concentration of negative charges in this experiment is around 1.1 to 1.4 x 10-5 moles (corresponding to 33 g/L of polymer solution), and the amount of Ca2+ necessary to form the gel ranges from 1.5 to 2.5 x 10-5 moles. However, this is at odds with the statement that "The presence of precipitate is then observed when the moles of salt is greater than the moles of available sulfonate functions", since the amount of salt is always higher than that of sulfphonate groups. Please modify your statement or explain in more detail.

5.- Page 7, lines 216-217. It is stated that "the molar concentration of Ca2+ must be equal to 0.2 M to obtain the gel formation". First of all, if this refers to the experimental procedure explained in Page 3, lines 119-121, the final concentration of calcium in the reaction mixture is 1/6 of 0.2 M, because 50 microlitres of the 0.2 M solution are added to a final volume of 300 microlitres. Perhaps the authors meant the *starting* molar concentration of calcium. In this sense, how did the increase in Ca2+ was carried out in the experiment shown in Fig. 6A? Adding more volume from the same starting solution of Ca2+, or adding the same volume but from starting solutions of increasing concentrations?

6.- No gel was generated with the gallic acid derivative, but which were the conditions they attempted before to reach this conclusion?

7.- Page 7, line 220. It is said that, for the tannic acid derivative case, a minimum amount of 1.17 x 10-5 moles of sulphonate groups are needed to form a gel. However, Fig. 6B seems to indicate a higher number of these sulphonate groups.

Author Response

Dear reviewer,

Thank you for your positive comments and we agree with yours comments. We modified our manuscript.

1.- The presentation of results is, overall, very confusing, and it takes a long time and effort for the reader to get the necessary insight. For instance, the molecular mass of the sulphonated polymer is not given. This makes it very difficult to translate concentrations in g/L into moles of negative charges. An explanation on this procedure should be detailed. Moreover, some quantities are shown in moles, other in g/L, grams... while all these are useful to be maintained, for instance to easily reproduce the experiments, at least the number of moles should be always added to the text as well.

We precise the molar mass of the polymers and the number of moles in the text and in Table 1.

2.- The Experimental Procedures section does not always reflect the experiments shown in the text (e.g. Figs 5 or 6), but only one condition for gel formation is indicated.

We only precise one condition in the experimental part and in the text we precise the conditions used to obtain Figure 5. We added increasing volumes of positive charge solution (Ca2+ or TA-N(CH3)3+) from the same initial solution.The initial volume of the polymer in solution is 250 microliters.

3.- The degree of substitution of tannic acid by amine groups is said to be 88% and corresponding to 14 nitrogens with positive charge. It would be useful to explain in more detail the derivatization reaction and subsequent calculations made by the authors. For instance, upon inspection of Figure 3, it might be thought that only those OH groups in the "terminal" aryl rings are able to react with the epoxyde. If this were the case, there are 15 of these groups (3 per each terminal ring), of which the 88% accounts for 13, not 14 nitrogens added. Also, this would mean that the rest of OH groups are not reactive. Is this anything that has been described before?

The number of ammonium groups N(CH3)3+ was calculated from the area ratio of the peaks at 6.7-7.5 ppm ( I1,2), which corresponds to the protons on the aromatic group of tannic acid, and the peak at 3.2 ppm (Id), characteristic of the protons on the ammonium groups (equation 1). The conversion of the grafting reaction is defined by the equation 2. These equations are presented in the revised version. The substitution degree of TA was calculated as around 76% (that corresponds to 19 N+ per molecule). 

There is a mistake in our calculations and we thank the reviewer for its very helpful comment.

4.- Pages 6-7, lines 203-209, and Figure 5. I have estimated that the concentration of negative charges in this experiment is around 1.1 to 1.4 x 10-5 moles (corresponding to 33 g/L of polymer solution), and the amount of Ca2+ necessary to form the gel ranges from 1.5 to 2.5 x 10-5 moles. However, this is at odds with the statement that "The presence of precipitate is then observed when the moles of salt is greater than the moles of available sulfonate functions", since the amount of salt is always higher than that of sulfonate groups. Please modify your statement or explain in more detail.

We modified our explanation because the text was not clear. 

5.- Page 7, lines 216-217. It is stated that "the molar concentration of Ca2+ must be equal to 0.2 M to obtain the gel formation". First of all, if this refers to the experimental procedure explained in Page 3, lines 119-121, the final concentration of calcium in the reaction mixture is 1/6 of 0.2 M, because 50 microlitres of the 0.2 M solution are added to a final volume of 300 microlitres. Perhaps the authors meant the *starting* molar concentration of calcium. In this sense, how did the increase in Ca2+ was carried out in the experiment shown in Fig. 6A? Adding more volume from the same starting solution of Ca2+, or adding the same volume but from starting solutions of increasing concentrations?

We explain more in details the experimental section. To do this study we added increasing volumes of positive charge solution (Ca2+ or TA-N(CH3)3+) from the same initial solution. We precise the number of moles in Table 1.

6.- No gel was generated with the gallic acid derivative, but which were the conditions they attempted before to reach this conclusion?

We used the same conditions that we used in the case of Tannic acid derivative and Ca2+(Figure 6). The number of positive charges is inferior to 5.1.10-5 moles and number of sulfonate is inferior to 5.1.10-5 moles.

7.- Page 7, line 220. It is said that, for the tannic acid derivative case, a minimum amount of 1.17 x 10-5 moles of sulphonate groups are needed to form a gel. However, Fig. 6B seems to indicate a higher number of these sulfonate groups.

We modified our manuscript to improve this point. In figure 5  the concentration of PHOSO3- is 27 g.L-1 ( 1.1. 10-5 moles). In the Figure 6, we tested different concentrations of PHOSO3- (from 0 to 5.1 10-5 moles). As a consequence we showed that wit is possible to obtain a gel when the number of sulfonate groups is equal 0.483.10-5 moles.

Reviewer 4 Report

In this article, new gel materials based on mcl-PHA have been generated by the synthesis of an intermediate PHA sulfonate and the subsequent elaboration of a crosslinked structure by its combination with bivalent cations or a tannic acid derivative molecule. The viscoelastic properties of the gels and the antioxidant activity of the one with best mechanical properties were studied.

Some points should be considered before publication:

1.- Abstract: the authors should confirm if these new material would still be renewable after the introduction of sulfonate groups and the tannic acid derivative. Otherwise this term should be removed until it proves right.

2.- Line 67: please, include some interesting properties of bivalent cations in the text.

3.- The method described in the M&M section to elaborate the PHOSO3- based networks is very precise about the amount of each component used. However, in the Results section it seems as if those amounts are the best ones to obtain the gel, but many other were tested (e.g. Figs. 5 and 6). The M&M section should include the experiments exactly as they were performed and presented in the Results section.

4.- Please indicate the MW of the monomeric unit of the polymer.

5.- Line 89: the PHOU characterization after synthesis should be included. How do the authors know that the polymer includes 33 % of double bonds?.

6.- Error bars should be included in all the figures and throughout the text, and the number of repetitions performed should be stated.

7.- Lines 212-225: there is a completely mess in the text between concentration measures. The authors mix moles, g L-1 and M when explaining the gel formation. The units should be shown uniformly to allow a better comparison.

8.- Fig. 5: the concentration of PHOSO3- indicated is 33 g L-1 whereas in M&M they mention 27 g L-1.

9.- Fig. 6 is not properly explained and, in fact, there is not concordance between the data in the text and in the figure. For example, in Fig. 6A, gel formation seems to appear from more than 2 * 10-5 moles of SO3-, whereas in the text (line 220) the authors mention 1.17 * 10-5 moles.

10.- Table 1: please, indicate that the mechanical properties shown are for the PHOSO3-/Ca2+ network.

11.- Line 252: the authors say that elastic modulus of the PHOSO3-/TA-N(CH3)3+ network reaches 4700 Pa, whereas in the abstract they mention 4200 Pa. Please, correct it.

12.- English grammar and wording should be carefully revised.

Author Response

Dear editor, 

Thank you for yours comments that help us to improve the quality of our manuscript. 

1.- Abstract: the authors should confirm if these new material would still be renewable after the introduction of sulfonate groups and the tannic acid derivative. Otherwise this term should be removed until it proves right.

This term is removed

2.- Line 67: please, include some interesting properties of bivalent cations in the text.

Among the bivalent cations which have the capability of forming gel by ionic interactions with anionic polymers, Ca2+, Ba2+ and Mg2+ present interesting properties in biomedical applications [37-39]. In particular, the gelation of an alginate solution with a multivalent ion of opposite charge such as Ca2+ allows to obtain resistant and biocompatible hydrogel.

3.- The method described in the M&M section to elaborate the PHOSO3- based networks is very precise about the amount of each component used. However, in the Results section it seems as if those amounts are the best ones to obtain the gel, but many other were tested (e.g. Figs. 5 and 6). The M&M section should include the experiments exactly as they were performed and presented in the Results section.

We only precise one condition in the experimental part and in the text we precise the conditions used to obtain Figure 5. We added increasing volumes of positive charge solution (Ca2+ or TA-N(CH3)3+) from the same initial solution.The initial volume of the polymer in solution is 250 microliters.

4.- Please indicate the MW of the monomeric unit of the polymer.

The molar mass of the monomeric unit of the polymer is M0 = 0.67*142 + 0.33*323 = 201 g.mol-1. It will be indicated in the revised manuscript (experimental part) 

6.- Line 89: the PHOU characterization after synthesis should be included. How do the authors know that the polymer includes 33 % of double bonds? 

PHOU was first characterized by 1H NMR (Figure 2) to determine the percentage of terminal unsaturation by integrating protons corresponding to the CH peak (2) at 5.1 ppm, and the signal relating to the terminal alkene group of side chain (7) at 5.7 ppm.  

7.- Lines 212-225: there is a completely mess in the text between concentration measures. The authors mix moles, g L-1 and M when explaining the gel formation. The units should be shown uniformly to allow a better comparison.

We improved our manuscript 

8.- Fig. 5: the concentration of PHOSO3- indicated is 33 g L-1 whereas in M&M they mention 27 g L-1.

Figure 5 is obtained with 27g.L-1. In M&M we mention only one example. 

9.- Fig. 6 is not properly explained and, in fact, there is not concordance between the data in the text and in the figure. For example, in Fig. 6A, gel formation seems to appear from more than 2 * 10-5 moles of SO3-, whereas in the text (line 220) the authors mention 1.17 * 10-5 moles.

We corrected our manuscript

10.- Table 1: please, indicate that the mechanical properties shown are for the PHOSO3-/Ca2+ network.

We indicated it.

11.- Line 252: the authors say that elastic modulus of the PHOSO3-/TA-N(CH3)3+ network reaches 4700 Pa, whereas in the abstract they mention 4200 Pa. Please, correct it.

We corrected it.

12.- English grammar and wording should be carefully revised.

We revised the english language.